# Double-Pigtail Drainage Catheter: A New Design for Efficient Pleural Drainage

**DOI:** 10.3390/medicina59061089

**Published:** 2023-06-05

**Authors:** Youngjong Cho, Hyoung Nam Lee, Ji Hoon Shin, Sung-Joon Park, Sangjoon Lee, Jae-Seok Song

**Affiliations:** 1Department of Radiology, University of Ulsan College of Medicine, Gangneung Asan Hospital, 38 Bangdong-gil, Gangneung 25440, Republic of Korea; ohggamja@gmail.com; 2Department of Radiology, Soonchunhyang University College of Medicine, Cheonan Hospital, Cheonan 31151, Republic of Korea; radiology2010.hnl@gmail.com; 3Department of Radiology, University of Ulsan College of Medicine, Asan Medical Centre, 88 Olympic-ro 43-gil, Seoul 05505, Republic of Korea; jhshin@amc.seoul.kr; 4Department of Radiology, Korea University College of Medicine, Korea University Ansan Hospital, Ansan 15355, Republic of Korea; dpcultintv@korea.ac.kr; 5Vascular Center, The Eutteum Orthopedic Surgery Hospital, Paju 10905, Republic of Korea; wannafindlee@gmail.com; 6Department of Preventive Medicine & Public Health, Catholic Kwandong University College of Medicine, 24 Beomil-ro 579beon-gil, Gangneung 25601, Republic of Korea

**Keywords:** pigtail catheter, pleural drainage, simple pleural effusion

## Abstract

*Background and Objectives*: The novel double-pigtail catheter (DPC) has an additional pigtail coiling at the mid-shaft with multiple centripetal side holes. The present study aimed to investigate the advantages and efficacy of DPC in overcoming the complications of conventional single-pigtail catheters (SPC) used to drain pleural effusion. *Materials and Methods*: Between July 2018 and December 2019, 382 pleural effusion drainage procedures were reviewed retrospectively (DPC, *n* = 156; SPC without multiple side holes, *n* = 110; SPC with multiple side holes (SPC + M), *n* = 116). All patients showed shifting pleural effusions in the decubitus view of the chest radiography. All catheters were 10.2 Fr in diameter. One interventional radiologist performed all procedures and used the same anchoring technique. Complications (dysfunctional retraction, complete dislodgement, blockage, and atraumatic pneumothorax) were compared among the catheters using chi-square and Fisher’s exact tests. Clinical success was defined as an improvement in pleural effusion within three days without additional procedures. Survival analysis was performed to calculate the indwelling time. *Results*: The dysfunctional retraction rate of DPC was significantly lower than that of the other catheters (*p* < 0.001). Complete dislodgement did not occur in any of the DPC cases. The clinical success rate of DPC (90.1%) was the highest. The estimated indwelling times were nine (95% confidence interval (CI): 7.3–10.7), eight (95% CI: 6.6–9.4), and seven (95% CI: 6.3–7.7) days for SPC, SPC + M, and DPC, respectively, with DPC showing a significant difference (*p* < 0.05). *Conclusions*: DPC had a lower dysfunctional retraction rate compared to conventional drainage catheters. Furthermore, DPC was efficient for pleural effusion drainage with a shorter indwelling time.

## 1. Introduction

Recently, pigtail catheter insertion has become the preferred alternative minimally invasive treatment over conventional large-bore chest tubes for pleural effusion drainage. The pigtail catheter in the pleural space has demonstrated its effectiveness in various indications, including simple effusion and pneumothoraxes. It is now widely accepted due to its proven benefits and positive outcomes [1,2,3]. Placing a large-bore chest tube is relatively invasive, with potential morbidities and complications [4]. Pigtail catheters inserted into the pleural space have a smaller profile, with more flexible materials causing less pain and injury to surrounding tissues, including intercostal vessels and nerves [5]. Fewer pigtail catheter complications were documented in uncomplicated pleural effusions compared to chest tubes [6]. Furthermore, additional treatment, such as pleurodesis or fibrinolysis, can be performed using pre-existing catheters [7].

Maintaining the proper position of an indwelling catheter is important for efficient drainage. Inserted catheters can be unintentionally pulled out. Nevertheless, catheters can be securely fixated onto the skin by thoroughly educating the patients. An improper position can compromise catheter function and, occasionally, dislodge catheters completely.

The novel double-pigtail catheter (DPC), which has an additional pigtail coiling in the middle of the catheter, was introduced to overcome these drawbacks. The additional pigtail coiling is believed to prevent complete dislodgement and maintain catheter function during the indwelling period by resisting retraction. Therefore, the present study aimed to compare the complication and clinical success rates of conventional single-pigtail catheters (SPCs) without multiple side holes, SPCs with multiple side holes (SPC + M), and DPCs for pleural effusion drainage.

## 2. Materials and Methods

### 2.1. Study Population

The requirement to obtain informed consent from participants was waived due to the retrospective study design. The study’s protocol was approved by the Institutional Review Board of Gangneung Asan Hospital (approval no.: GNAH 2020-12-006). Between July 2018 and December 2019, we retrospectively reviewed all consecutive drainage procedures for the initially inoculated pleural effusion of varying etiologies. Cases where the loculation was evident from the start or when repositioning was requested after a previously inserted pigtail catheter for pleural effusion were excluded from the study. A shifting pleural fluid collection was verified in the decubitus view of simple chest radiographs conducted as the baseline examination for all patients. A total of 382 pleural drainage procedures were performed in 343 patients. The ECOG (European Cooperative Oncology Group) performance status, number of comorbidities, and potential cause of pleural effusion in these patients were investigated using medical records. Additionally, fluid analysis following the use of the chest pigtail catheter was performed, distinguishing transudate and exudate according to the Light’s criteria.

### 2.2. Types of Percutaneous Drainage Pigtail Catheters

The pigtail catheters available for pleural drainage were as follows: (1) SPC: Dawson–Mueller Drainage Catheter, made of Ultrathane, with a hydrophilic coated distal segment (5 cm, five side holes with a pigtail formation of 10 mm) and Mac–Loc for the suture fixation system (25 cm in delivery length and 0.038” inner diameter; Cook, Bloomington, IN, USA); (2) SPC + M: Biliary Drainage Catheter, made of Ultrathane, with a hydrophilic coated distal segment (12 cm, 9 side holes with pigtail formation) and side-port segment (8 cm, 22 side holes), and Mac–Loc for the suture fixation system (40 cm in delivery length, 0.038” in the inner diameter; Cook, Bloomington, IN, USA); (3) DPC: Multi-Purpose Drainage Catheter, made of polyurethane, double-loop pigtail, and a suture fixation system, with a total of 21 side holes (5 for each pigtail and 11 side-port segments), pigtail loop size of 2 cm with 7 cm intervals, delivery length of 30 cm, and 0.038” in the inner diameter (Sungwon Medical, Cheongju, North Chungcheong Province, Republic of Korea; Figure 1a,b).

### 2.3. Procedure Details

One intervention radiologist (BLINDED-FOR-REVIEW) performed all procedures in the intervention radiology suite with the patients in a supine position. Combined fluoroscopy and ultrasound were used to guide the drainage catheter’s placement using the Seldinger technique as follows: (1) The hypoechoic fluid collection was confirmed on ultrasonography in the intercostal space of the lower ribs, approximately at the level of the mid-axillary line. (2) For local anesthesia, 2% lidocaine was administered at the puncture site, and the location of the needle (21G Chiba needle) was continuously verified using an ultrasound probe, with the needle “walking over” the upper edge of the rib to avoid intercostal neurovascular bundles. (3) After puncturing the pleura, a metallic hair wire with a floppy tip (0.018”) was inserted through the needle. A 6-Fr dilator sheath was inserted (3S-Percutaneous access kit, Dukwoo Medical, Hwaseong, Gyeonggi, Republic of Korea). (4) Through the sheath, a 0.035” hydrophilic guide wire (Radifocus, Terumo, Tokyo, Japan) was inserted. (5) Under fluoroscopic guidance, 10.2-Fr pigtail catheters were inserted over the guidewire. The catheter’s tip was located at the T6–8 level, which is presumed to be the most dependent portion of fluid collection in the supine position. (6) One interventional radiologist applied the same anchoring technique in all patients (using five knots; one loop of thread around the tubing between the knots of 2-0 black silk).

### 2.4. Technical and Clinical Success and Complications

Technical success was defined as smooth drainage immediately after the successful installation of the catheter in the thoracic cavity with pleural effusion. Clinical success was defined as an improvement in pleural effusion upon follow-up simple chest radiography within 3 days without additional procedures. The indwelling time was defined as the catheterization period, and when the catheter was removed or changed because of complications, it was treated as censored data. Major complications included dysfunctional retraction, complete dislodgement, and blockage. Dysfunctional retraction was defined as the outward pulling of the distal pigtail loop of the catheter without leaving the pleural cavity, causing dysfunction in the drainage and replacement of the catheter (Figure 2a,b). Complete dislodgement was defined as the dislocation of the catheter end outside the pleural cavity where drainage fluid is located (Figure 2c). Blockage was defined as a rapid decrease in daily drainage with a significant amount of pleural effusion remaining in simple chest radiographs. Other complications, such as atraumatic pneumothorax (Figure 2b), hemorrhage, and subcutaneous emphysema (Figure 2a), were recorded for each catheter during the indwelling time. A pneumothorax refers to the presence of air in the pleural space, which is the space between the lungs and the chest wall. It can occur due to various reasons, including trauma or injury to the chest, lung diseases, or medical procedures. However, the reason for specifying it as an atraumatic pneumothorax was that the air accumulation in the pleural space was not caused by injury from the catheter but rather occurred independently due to various factors during the indwelling period.

### 2.5. Statistical Analysis

Categorical variables including demographical data and other baseline characteristics were compared using chi-square and Fisher’s exact tests. Complications of each catheter were also compared using chi-square and Fisher’s exact tests. When the *p*-value among the three groups was found to be significant, additional comparisons were conducted between the two specific groups. Kaplan–Meier survival analysis was employed to calculate the indwelling time of each catheter. A Log-rank test was performed for comparing the indwelling time of each catheter in clinical success cases. Statistical analyses were performed using SPSS version 25.0 (IBM, Armonk, NY, USA) and Lifelines Python package (version 0.26.0, Davidson-Pilon C., DOI 10.5281/zenodo.805993) were used and *p*-values < 0.05 were considered to indicate statistical significance.

## 3. Results

Demographic data did not differ significantly among the groups. No significant differences were observed among the three groups in terms of ECOG performance status, number of comorbidities and etiology of pleural effusion. There was no statistically significant difference observed in the composition of transudate and exudate according to Light’s criteria in the pleural fluid analysis (Table 1). All procedures were technically successful without immediate complications, including bleeding and traumatic pneumothorax. The clinical success rate of the SPC, SPC + M, and DPC groups were 83.7%, 81.2%, and 90.1%, respectively, showing significant differences (*p* = 0.029).

Regarding complications, the rates of dysfunctional retraction in the DPC, SPC + M, and SPC groups were 4.5%, 15.5%, and 23.6%, respectively (*p* < 0.001), showing significant differences (Table 2). When comparing the two groups, a statistically significant difference was observed, especially when comparing SPC + M and DPC, as well as SPC and DPC.

There were three cases of complete dislodgement in the entire study population: two in the SPC group and one in the SPC + M group. One SPC + M catheter was continuously retracted (Figure 2a) and wholly dislodged from the pleural cavity. In the other two cases, patients unconsciously pulled out their catheters at night, and the catheters were missing in follow-up radiographs. However, complete dislodgement did not occur in the DPC group.

The frequencies of blockage in the SPC + M, DPC, and SPC groups were nine (8.2%), eight (5.1%), and four (3.4%), respectively, showing no statistical significance.

As for other complications, atraumatic pneumothorax was the most common (*n* = 16: 6 in the SPC group, 5 in the SPC + M group, and 5 in the DPC group). Chest wall hemorrhage (*n* = 1) and subcutaneous emphysema (*n* = 1) were also detected in the SPC + M group (Table 2).

In the Kaplan–Meier survival analysis, the estimated indwelling times were eight (95% confidence interval (CI): 7.3–10.7), nine (95% CI: 6.6–9.4), and seven (95% CI: 6.3–7.7) days for SPC, SPC + M, and DPC, respectively. For DPC, it was confirmed that the indwelling period was statistically significantly shorter when compared to SP and SP + M (*p* < 0.05) (Figure 3a,b).

## 4. Discussion

In our study, we specifically targeted non-loculated pleural effusion cases without septations, as confirmed by intra-procedural ultrasound. This decision was made because, for septated pleural effusion, exudate, or hemorrhage, a chest tube with a larger diameter may result in more effective drainage. Consequently, we compared the efficacy and safety of pigtail catheters for relatively straightforward pleural effusion cases.

Pigtail catheter insertion is a minimally invasive procedure where a small, flexible tube with a coiled end is placed into the pleural space. It is a less invasive alternative to chest tube insertion and is commonly performed using local anesthesia. For uncomplicated effusions, pigtail catheters ranging from 8Fr. to 14Fr. are suitable, offering lower risks of complications and shorter hospital stays. However, there can still be issues such as catheter dysfunction and blockages. Dysfunctional retraction and complete dislodgement are common complications following percutaneously inserted drainage procedures [8]. The rates of these complications following pleural drainage procedures have rarely been reported [1,5,6]. The drainage catheters may become malpositioned or gradually pull away because of respiratory motion or improper management, including physical traction. Maintaining their optimal position is crucial in order to maintain efficient draining capabilities during indwelling times [9]. It can decrease patient discomfort and reduce associated medical costs. To place the side holes of a catheter into the intended space, additional pigtail coiling was formed at the mid-shaft of the drainage pigtail catheter. These catheters’ performances in preventing retraction or dislodgement have not been studied before.

Dysfunctional retraction was most frequently noted in the SPC + M group, followed by SPC and DPC groups. In cases of SPC + M, catheter dysfunction was more likely to occur when some of the side holes at the mid-shaft were pulled outside the pleural cavity (Figure 2a). These pulled-out side holes can be blocked inside the chest wall and cause dysfunction, resulting in replacement. In addition, SPC can become dysfunctional when the pigtail loop is fully retracted and hangs onto the parietal pleura (Figure 2b). In contrast, when unintended traction occurred in DPC, additional pigtail coiling at the mid-shaft of the catheter prevented further retraction. The function of DPC was preserved by limiting the range of retraction (Figure 4a,b). Furthermore, while retraction of the midshaft coiling may occur, complete dislodgement of DPC was not observed.

After sufficient drainage of pleural effusion, air can enter the pleural space if the lung is not adequately expanded due to lung trapping. Alternatively, air can accumulate in the pleural space in a check-valve manner as the catheter is retracted or partially obstructed. Identifying these causes accurately can be challenging, so they were considered as one of the complications due to their association with catheter dysfunction. Therefore, in this study, the air collection in the pleural space was described as an atraumatic pneumothorax to distinguish it from the traumatic pneumothorax that occurred during the pigtail catheter’s insertion. These atraumatic pneumothoraxes occurred with all three types of catheters, and they are possibly associated with negative intrapleural pressure during inspiration. However, this mainly occurred in the indwelling period and could be managed if the catheter’s dysfunction had not occurred.

Cases of subcutaneous emphysema and chest wall hemorrhage were observed exclusively in the SPC catheter. Subcutaneous emphysema is believed to be a complication resulting from the presence of multiple side holes in the shaft of the catheter, originally intended for efficient drainage. The positioning of these side holes in the chest wall during retraction is considered to be problematic (Figure 2a). Since subcutaneous emphysema caused great discomfort to the patient, it is regarded as a major complication. However, in the case of chest wall hemorrhage, it is believed to be caused by inadvertent contact with blood vessels beneath the ribs. This occurrence is not considered to be specifically related to the type of catheter used. The fact that it was only observed in 1 out of 382 pleural drainage procedures suggests the effectiveness of using a small-bore pigtail catheter instead of a chest tube.

Pleurodesis was successfully performed using a 10.2 Fr. pigtail catheter in patients with malignant effusion. Specifically, SPC (*n* = 3), SPC + M (*n* = 1), and DPC (*n* = 1) procedures were carried out. The pleurodesis agent used was Abnoba Viscum or mistletoe extract, with a dosage of five ampoules containing 20 mg/1 mL in our hospital. Mistletoe extracts, including AbnobaViscum, have been suggested to have immunomodulatory and anti-tumor effects, which may contribute to their potential use in pleurodesis. The idea behind using mistletoe extract in pleurodesis is to induce an inflammatory response in the pleural space, leading to the formation of adhesions and subsequent fusion of the pleural layers. However, the effectiveness and safety of mistletoe extract for pleurodesis have not been thoroughly established, and further research is needed to evaluate its benefits and risks in this specific application.

Adequate drainage would be achieved if more side holes are well placed in the intended position. However, paradoxically, some debris can become stuck in the side hole, impairing catheter patency. When replacing or removing the catheter, in the case of SPC + M, substantial debris was stuck in multiple side holes of the linear shaft, which was presumed to cause the blockage of the passage. Side holes inside the pigtail loop were presumed to be less vulnerable to blockage because they faced the centripetal direction inside the loop. Catheters with multiple side holes (SPC + M and DPC) showed a higher blockage rate than SPC. In addition, SPC + M showed a higher blockage rate than DPC, as SPC + M had more side holes on the shaft (22 vs. 11). However, these findings failed to demonstrate statistical significance. Therefore, it is believed that adding an additional pigtail in the midshaft without increasing the number of side holes can decrease overall complications. Taking this into consideration, further investigation using a newly designed double pigtail catheter with only side holes at the end pigtail loop may be warranted.

The clinical success rates of these three types of catheter are comparable to those in a previously published study [10]. Regardless of the type, the pleural drainage pigtail catheter demonstrated a high clinical efficacy rate of over 80%. As mentioned earlier, the SPC + M group showed the lowest clinical success rate among all types of catheters. This may be because SPC + M was more vulnerable to both retraction and blockage. DPC resisted dysfunctional retraction compared to other types of catheters and showed a comparable blockage rate to SPC. It can be inferred that preventing these catheter complications ultimately leads to improved clinical efficacy. In addition, DPC showed a significantly shorter indwelling time than other types of catheters according to Kaplan–Meier survival analysis of the catheters. The longer proper positioning of the catheter seemed to decrease the indwelling time. This can help to minimize patient discomfort and shorten the length of hospital stays reducing medical costs.

This study has some limitations. First, this study had a retrospective design. Second, randomization was not performed in the catheter type selection. However, the baseline characteristics did not differ significantly among the three groups of catheters. Third, the amount or nature of the patient’s pleural effusion (viscosity, turbidity, and amount of debris) was not constant. However, it is important to note that the catheter anchoring technique and profile, which are crucial factors for catheter fixation, are also regulated at the 10.2 Fr. size. In addition, all procedures were performed by one interventional radiologist; therefore, only one anchoring technique was used. As the method of fixing the catheter to the skin was unified, comparing dysfunctional retraction or complete dislodgement under the same conditions was possible. The minimally invasive treatment using a drainage pigtail catheter has proven to be useful not only in pleural drainage but also in various fields such as the biliary system and urinary system. However, issues related to catheter retraction and dislodgement have also been observed in theses fields, which can impact patient treatment, cause patient discomfort, and increase unnecessary medical costs. Therefore, further research is necessary to broaden the design and application of double pigtail catheters in drainage systems across different medical fields.

## 5. Conclusions

DPC had a lower dysfunctional retraction rate compared to other conventional pleural drainage catheters because it limited the range of retraction. It also had a higher clinical success rate than SPC + M. Furthermore, it was efficient and exhibited a shorter indwelling time, possibly resulting in shortened hospital stays and reduced costs.

## Figures and Tables

**Figure 1 medicina-59-01089-f001:**
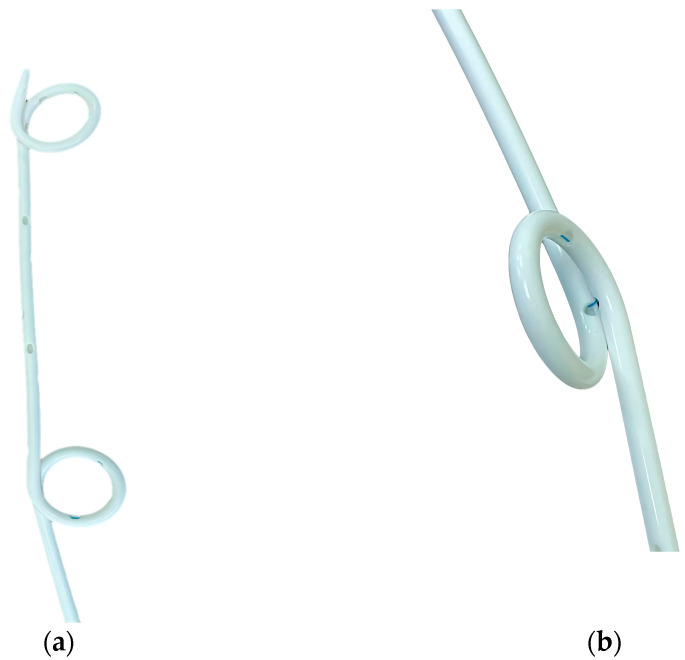
Detailed structure of a double-loop pigtail catheter. (**a**) Multi-Purpose Drainage Catheter, (**b**) consisting of a double-loop pigtail structure by adding another pigtail coiling in the middle of the shaft. Upon closer examination, it can be observed that the pigtail loop in the middle is a structure that can be secured using a thread.

**Figure 2 medicina-59-01089-f002:**
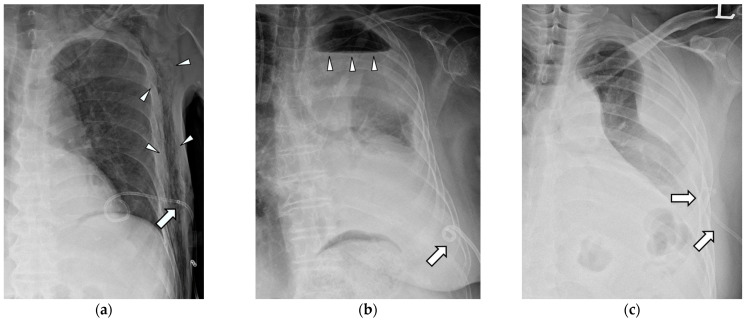
Complications of conventional percutaneous drainage pigtail catheters. (**a**) The drainage pigtail catheter was retracted, and a radio-opaque marker (arrow) indicating the locations of side holes was positioned at the chest wall level. This resulted in catheter dysfunction and subcutaneous emphysema (arrowheads). (**b**) The single-pigtail catheter without side holes was fully retracted to the chest wall. Hydropneumothorax (arrowheads) was not resolved because of the malfunctioning of the catheter. (**c**) Dislodgement occurred at the pigtail’s tip located completely outside the pleural cavity (arrows).

**Figure 3 medicina-59-01089-f003:**
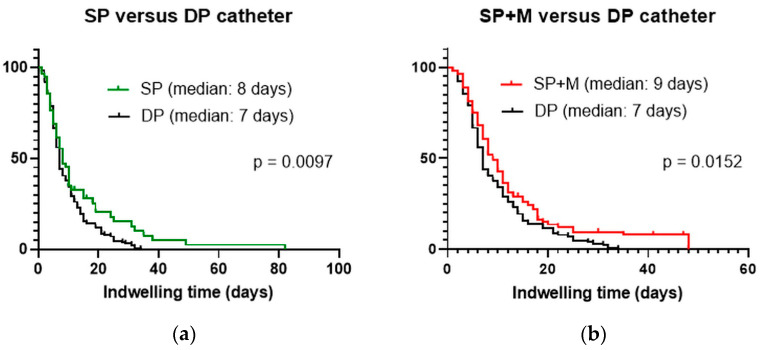
Kaplan–Meier estimates of the indwelling time for percutaneous pleural drainage catheters. (**a**) Single-pigtail catheter versus double-pigtail catheter. (**b**) Single-pigtail catheter with multiple side-holes versus double-loop pigtail catheter.

**Figure 4 medicina-59-01089-f004:**
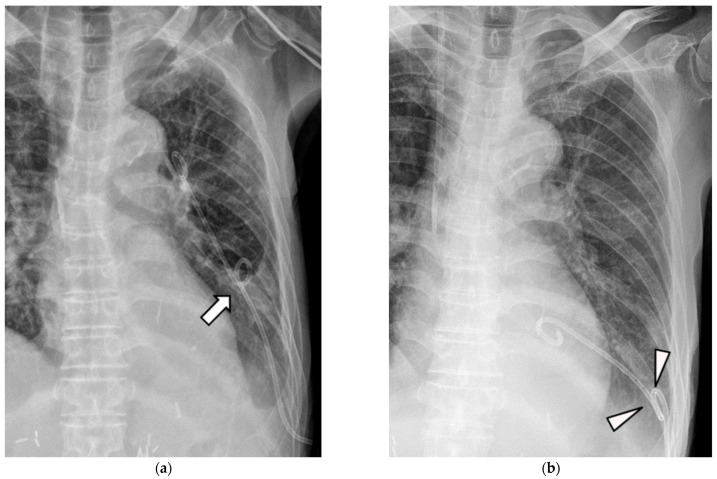
A 67-year-old man with pleural effusion treated with a double-pigtail catheter. (**a**) The double-loop pigtail catheter was inserted for left pleural effusion. Note that the second pigtail loop was at the middle lung field. (**b**) The catheter was mildly retracted to the site where the second pigtail loop hung onto the pleura (arrowheads). The function of the catheter was maintained despite mild retraction.

**Table 1 medicina-59-01089-t001:** Summary of patients’ baseline characteristics.

	SPC (116)	SPC + M (110)	DPC (156)	*p*-Value
**No. of patients**	108	99	136	
**Age (years)**, mean ± SD	71.7 ± 12.7	72.6 ± 12.0	70.1 ± 13.5	0.407
**Sex (No.)**				0.683
Male	70	59	87	
Female	38	40	48	
**ECOG, PS (No.)**				0.501
0	4	6	6	
1	36	21	43	
2	51	59	74	
3	14	11	10	
4	3	2	3	
**No. of comorbidities** (median and IQR)	2, (1–4)	2, (1–4)	2, (1–4)	0.879
**Etiology (No.)**				0.976
Cardiac/renal failure	38	40	56	
Parapneumonic effusion	22	14	22	
Malignant effusion	30	26	32	
Tbc pleurisy	12	11	15	
Liver cirrhosis	2	3	4	
Postoperative	2	2	5	
Pneumoconiosis	2	3	2	
**Pleural fluid characteristics**				0.366
Transudate (No., %)	44 (40.7%)	48 (48.5%)	67 (49.3%)	
Exudate (No., %)	64 (59.3%)	51 (51.5%)	69 (50.7%)	

SPC—Single-pigtail catheter; SPC + M—single-pigtail catheter with multiple side holes; DPC—double-loop pigtail catheter; SD—standard deviation; ECOG—Eastern Cooperative Oncology Group; PS—performance status; IQR—interquartile range; Tbc—tuberculosis.

**Table 2 medicina-59-01089-t002:** Summary of complications of each catheter and comparison.

	SPC (116)	SPC + M (110)	DPC (156)	SPC vs. SPC + M	SPC + M vs. DPC	SPC vs. DPC
**Dysfunctional retraction**	18 (15.5%)	26 (23.6%)	7(4.5%)	*p* = 0.123	*p* < 0.001	*p* < 0.001
**Complete dislodgement**	2 (1.7%)	1 (0.9%)	0 (0%)	*p* = 1	*p* = 0.41	*p* = 0.41
**Blockage**	4 (3.6%)	9 (8.2%)	8 (5.1%)	*p* = 0.158	*p* = 0.316	*p* = 0.445
**Other**	6 (5.2%)	7 (6.4%)	5 (3.2%)	*p* = 0.701	*p* = 0.222	*p* = 0.415
Atraumatic pneumothorax	6	5	5			
Chest wall hemorrhage	0	1	0			
Subcutaneous emphysema	0	1	0			

SPC—Single-pigtail catheter; SPC + M—single-pigtail catheter with multiple side holes; DPC—double-loop pigtail catheter; N/A—not available.

## Data Availability

The dataset generated and/or analyzed during the current study is available from the corresponding author upon reasonable request.

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
