# Peer review of "Double-Pigtail Drainage Catheter: A New Design for Efficient Pleural Drainage"

_medicina, 2023, doi:10.3390/medicina59061089_

Round 1

Reviewer 1 Report

Need major revision 

Need major revision 

Author Response

Response to Reviewer 1 Comments

Point 1: (Abstract)

centripetal multiple -> multiple centripetal

The aim of the present study was to -> The present study aimed to

3 days -> three days

A survival analysis was performed for calculating -> Survival analysis was performed to calculate

DPC were -> DPC was

Response 1: Thank you for your comments and corrections. We edited abstract as you suggested.

Point 2: (Introduction)

insertion is -> insertion has been

over the conventional -> over conventional

fewer complications of pigtail catheters were documented in cases of simple pleural effusions compared to chest tubes -> Fewer pigtail catheter complications were documented in uncomplicated pleural effusions compared to chest tubes

using the pre-existing -> using pre-existing

important -> essential

To overcome these drawbacks, the novel double pigtail catheter (DPC), which has additional pigtail coiling in the middle of the catheter, was introduced. -> The novel double-pigtail catheter (DPC) which has additional pigtail coiling in the middle of the catheter, was introduced to overcome these drawbacks.

The aim of the present study was -> The present study aimed

Response 2: Thank you for your comments and corrections. We edited introduction as you suggested.

Point 3: (M&M)

All procedures were performed by one intervention radiologist (BLINDED-FOR-REVIEW) in the intervention radiology suite with the patients in the supine position. -> One intervention radiologist (BLINDED-FOR-REVIEW) performed all procedures in the intervetion radiology suite with the patients in the supine position.

,hypoechoic -> the hypoechoic

Deleting “ as follow ”

,and the -> .The

The same anchoring technique was applied in all patients by one interventional radiologist (number of knots: five; one loop of thread around the tubing between the knots of 2-0 black silk) -> One interventional radiologist applied the same anchoring technique in all patients (number of knots: five; one loop of thread around the tubing between the knots of 2-0 black silk).

after -> after the

Dysfunctional -> The dysfunctional

causing a dysfunction -> causing dysfunction

as dislocation -> as the dislocation

as rapid decrease -> as a rapid decrease

deleting “the amount of”

(Fig 2a) -> (Fig 2a),

A survival analysis -> Survival analysis

The clinical success 139 rate of SPC, SPC+M and DPC groups were 83.7%, 81.2% and 90.1% -> The clinical success rate of the SPC, SPC+M, and DPC groups were 83.7%, 81.2%, and 90.1%,

Response 3: Thank you for your comments and corrections. We edited M&M as you suggested.

Point 4: (Results)

The clinical success 139 rate of SPC, SPC+M and DPC groups were 83.7%, 81.2% and 90.1% -> The clinical success rate of the SPC, SPC+M, and DPC groups were 83.7%, 81.2%, and 90.1%,

The entire study -> the study

Deleting “and”

Eventually completely -> wholly

Response 4: Thank you for your comments and corrections. We edited results as you suggested.

Point 5: (Discussion)

Deleting “relatively”

Inappropriate -> improper

Maintaining 173 their optimal position is crucial to keep efficient draining capability during the indwelling 174 period [9]. -> Maintaining their optimal position is crucial to maintain efficient draining capability during indwelling [9]

Dysfunctional -> The dysfunctional

Dysfunction of the catheter -> catheter dysfunction

Side-holes -> side holes

SPC -> In addition, SPC

It mostly occurred in the indwelling 196 period and could be managed if the dysfunction of the catheter had not occurred. -> However, it mainly happened  in the indwelling period and could be managed if the catheter's dysfunction had not occurred.

Caused -> causes

Effective -> Adequate

Are -> were

Which impairs -> which impairing

Presumed to be -> presumed

Compared to -> than

Longer proper positioning of the catheter seemed to decrease indwelling time. -> In addition, the longer proper positioning of the catheter seemed to decrease indwelling time.

Stay -> stays

Patients’ -> patient’s

Was possible -> as possible

Response 5: Thank you for your comments and corrections. We edited discussion as you suggested.

Point 6: (Conclusions)

Compared to -> Than

Stay -> stays

Response 6: Thank you for your comments and corrections. We edited conclusions as you suggested.

Reviewer 2 Report

Dear Editor and Authors,

I read and evaluated the manuscript titled “Double-pigtail Drainage Catheter: A New Design for Efficient  Pleural Drainage” by Dr. Cho and colleagues from the Gangneung Asan Hospital, University of Ulsan College of Medicine in Gangwon-do, South Korea.

In this retrospective, single institution analysis the authors present their experience in pleural effusion drainage utilizing the novel double-pigtail catheter (DPC). Over a six-month period they used this novel double pig tail catheter with multiple holes which I believe was nationally produced (i.e. in Korea) in 156 patients and compared it to 110 patients which had a single pig tail (SPT) catheter without extra holes and 116 patients which has a SPT catheter with multiple size holes. The authors were able to show that DPC catheters stayed in situ longer and had a larger (90%) success rate.

There are a number of issues I feel need addressing in regards to this work:

1.       The manuscript needs an English language proofreading. The language is adequate to understand but there are numerous expression mistakes that need correction.

2.       Table 1 with basic demographics should be moved in the results section.

3.        Table 2 is mislabeled. It is not demographic characteristics but complications!

4.       The description of the insertion procedure is quite detailed and unnecessary. As thoracic surgeons and pulmonologists we are well aware of the technique utilized!

5.       I am not sure that what the authors report as pneumothorax represents that or space left post drainage due to lung trapping! Can the authors clarify the reason behind pneumothorax development?

6.       Age and Sex are the only personal demographics reported without indicating things like performance status, smoking history, co-morbidities. Please include some of these!

7.       You don’t need to report the full biochemical analysis of the fluid. One can report exudate or transudate instead!!

8.       How many patients underwent pleurodesis via the catheter? What was the average size of the catheters? What medium was used for pleurodesis? Talk, bleumycic, ect!

In conclusion, this is an interesting, small little study which has some interest although it needs some corrections. One concern I have is if there is a conflict of interest of the authors regarding the locally produced catheter (I know they report there isn’t but please confirm!!). Thank you for asking me to review this work. Good luck to all.

Please see above. The manuscript does need language editing mainly for expression issues.

Author Response

Response to Reviewer 2 Comments

I read and evaluated the manuscript titled “Double-pigtail Drainage Catheter: A New Design for Efficient  Pleural Drainage” by Dr. Cho and colleagues from the Gangneung Asan Hospital, University of Ulsan College of Medicine in Gangwon-do, South Korea.

In this retrospective, single institution analysis the authors present their experience in pleural effusion drainage utilizing the novel double-pigtail catheter (DPC). Over a six-month period they used this novel double pig tail catheter with multiple holes which I believe was nationally produced (i.e. in Korea) in 156 patients and compared it to 110 patients which had a single pig tail (SPT) catheter without extra holes and 116 patients which has a SPT catheter with multiple size holes. The authors were able to show that DPC catheters stayed in situ longer and had a larger (90%) success rate.

There are a number of issues I feel need addressing in regards to this work:

  1. The manuscript needs an English language proofreading. The language is adequate to understand but there are numerous expression mistakes that need correction.

Response 1: Thank you for your comment. For English proofreading, I have used language editing service after reflecting all comments below.

  1. Table 1 with basic demographics should be moved in the results section.

Response 2: Thank you for your comment. Demographics has moved in the results section.

  1. Table 2 is mislabeled. It is not demographic characteristics but complications!

Response 3: Sorry for mislableing table 2. We have corrected the name of Table 2.

  1. The description of the insertion procedure is quite detailed and unnecessary. As thoracic surgeons and pulmonologists we are well aware of the technique utilized!

Response 4: Thank you for your comment. We, therefore, reduced the technique section as much as possible.

  1. I am not sure that what the authors report as pneumothorax represents that or space left post drainage due to lung trapping! Can the authors clarify the reason behind pneumothorax development?

Response 5: This is a very valuable comment. After the effusion is drained, lung trapping and negative pressure in the pleural space create an air collection in the pleural space, that is, a atraumatic pneumothorax. I wrote an additional comment on this in result section because it is not a traumatic pneumothorax.

  1. Age and Sex are the only personal demographics reported without indicating things like performance status, smoking history, co-morbidities. Please include some of these!

Response 6: Thank you for your comment. We added performance status, and co-morbidities on Table 1.

  1. You don’t need to report the full biochemical analysis of the fluid. One can report exudate or transudate instead!!

Response 7: Thank you for your comment. Fluid analysis section was simplified, only dividing exudate or transudate based on Light’s Criteria.

  1. How many patients underwent pleurodesis via the catheter? What was the average size of the catheters? What medium was used for pleurodesis? Talk, bleumycic, ect!

Response 8: Thank you for your comment. We added a new paragraph about pleurodesis in discussion section.

In conclusion, this is an interesting, small little study which has some interest although it needs some corrections. One concern I have is if there is a conflict of interest of the authors regarding the locally produced catheter (I know they report there isn’t but please confirm!!). Thank you for asking me to review this work. Good luck to all.

It is a catheter that has already been allowed to be marketed in Korea, and we are not involved in catheter development at all, so there is no conflict of interest.

Round 2

Reviewer 1 Report

Reviewed manuscript is attended carefully and corrected the suggested comments

Reviewed manuscript is attended carefully and corrected the suggested comments

Reviewer 2 Report

Dear Editor and Authors,

I have re-read and re-evaluated the revised manuscript. It is better (not perfect I might say) but good enough for publication so I am recommending this now.

Take care,

Language needs some minor editing while undergoing proofreading.